# STRUCTURE-AWARE PRE-TRAINED LM CAN IMPROVE RNA SECONDARY STRUCTURE PREDICTION

## ABSTRACT

Accurate prediction of RNA secondary structure is a fundamental yet challenging task in computational biology, crucial for deciphering RNA's functional capabilities. While recent deep learning methods show promise, they are often limited by their failure to explicitly integrate structural information during pre-training and by the scale of their models and datasets. Here, we introduce Secondary Structure-Aware RNA Language Model(SSR-LM), a 650M-parameter language model pre-trained on 1.1 billion RNA sequences. A key innovation is our novel Secondary Structure-Aware Span Masking (SSM) pre-training task, which explicitly integrates structural motifs into the model. To address the lack of a comprehensive benchmark for evaluating model performance on real structures, we construct a new, large-scale PDB-derived RNA secondary structure dataset, three times larger than existing ones. Comprehensive evaluation demonstrates that SSR-LM achieves state-of-the-art performance, attaining an F1-score of $0.741$ on our new PDB benchmark and $0.635$ on the CASP16 blind test set, showcasing its robust performance.

## 1 INTRODUCTION

Ribonucleic acid (RNA) plays an important role in the central dogma of molecular biology, participating in diverse biological processes such as gene regulation, protein synthesis, and catalysis (Tinoco Jr & Bustamante, 1999). The secondary structure of RNA, defined by intramolecular base pairing via hydrogen bonds, gives rise to characteristic structural motifs including hairpins, loops, bulges, and pseudoknots (Tinoco Jr & Bustamante, 1999; Higgs, 2000). These structural elements are critical determinants of RNA stability, its interactions with proteins and other biomolecules, and ultimately, its functional roles within cellular processes (Mathews et al., 2010). Consequently, a thorough understanding of RNA secondary structure is vital for deciphering its diverse biological functions. However, the experimental determination of RNA structures can be time-consuming and resource-intensive, motivating the development of computational approaches (Deigan et al., 2009; Low & Weeks, 2010).

To address these challenges, researchers have increasingly turned their attention to computational methods, which offer faster and more scalable solutions, particularly for large or novel RNA sequences. Among these, thermodynamic modeling and deep learning have emerged as prominent strategies. Thermodynamic methods, such as those implemented in the widely used ViennaRNA package (Lorenz et al., 2011a), employ dynamic programming algorithms to predict the most stable structure by minimizing free energy based on established parameters. These methods are foundational for RNA structure prediction. However, for certain structured ncRNAs with intricate topologies such as complex pseudoknots, where deep learning has shown promise, these methods face challenges, motivating the development of complementary approaches. Deep learning methods, including convolutional neural networks(CNNs) (Sato et al., 2021; Fu et al., 2022) and recurrent neural networks(RNNs) (Singh et al., 2019), have demonstrated promise by learning complex patterns directly from RNA sequence data. However, their performance is frequently constrained by the quality and quantity of labeled training data.

Significant advancements in natural language processing (NLP) have established pre-trained Language Models (PLMs) as powerful tools for modeling RNA structure. The proliferation of high-throughput sequencing technologies has generated vast quantities of unlabeled RNA sequences, creating an ideal resource for training these models. While existing RNA language models, such as RNA-

FM (Chen et al., 2022), have demonstrated initial success in secondary structure prediction, there is still considerable room for improvement. A primary limitation is that many current models primarily rely on standard pre-training objectives like Masked Language Modeling (MLM) and just use ncRNA as pre-training data. While foundational, MLM by itself may not be sufficient for models to learn the complex structural information. This often stems from pre-training stages that do not sufficiently emphasize the explicit integration of such structural information. Although some approaches incorporate structural features (Yin et al., 2024), the strategy of deeply embedding structural motifs within the pre-training tasks themselves—a technique that has shown significant benefits in analogous protein modeling (Su et al., 2023; Hayes et al., 2025)—is not yet widely adopted for RNA. Furthermore, these models are sometimes constrained by smaller training datasets or limited model parameters (Chen et al., 2022; Penić et al., 2024a).

To overcome these limitations, we introduce **SSR-LM** (Secondary Structure-Aware RNA Language Model), a novel 650M RNA pre-trained language model. SSR-LM is pre-trained on an extensive dataset of **1.1 billion** RNA sequences and utilizes the LLaMA2 architecture (Touvron et al., 2023). To instill a deep understanding of RNA's structural principles, we introduce a novel structure-aware pre-training task called Secondary Structure-Aware Span Masking. In SSM, we mask a contiguous span of nucleotides within one strand of a structural element, such as an RNA stem. This task forces the model to move beyond local context; to accurately reconstruct the masked region, it must identify the complementary partner strand—often distant in the primary sequence—and apply the rules of base pairing. This objective effectively reframes self-supervised learning from simple token prediction to a puzzle of structural completion. Consequently, the model is compelled to learn representations that encode the long-range interactions governing RNA architecture, providing a powerful inductive bias for downstream structure-dependent tasks.

We conduct extensive evaluations to validate the effectiveness of our approach. These evaluations are performed on established public benchmark datasets for RNA secondary structure prediction. Furthermore, to enable more robust evaluation, we developed a larger benchmark dataset, derived from the Protein Data Bank (PDB). Furthermore, we extend our evaluations to MSA-free RNA tertiary structure prediction. Remarkably, SSR-LM exhibits exceptional accuracy in these challenging tasks, underscoring the significant value of pre-training with explicit secondary structure awareness. Our contributions can be summarized as follows:

- **Novel Pre-training Task**: We introduce SSM, a new pre-training task where the model masks important structural spans and learns to recover them. And we demonstrate the effectiveness of this approach through comprehensive experiments.

- **Large-Scale Structure-Aware Model**: We developed SSR-LM, a 650M parameter RNA PLM, pre-trained on an extensive dataset of 1.1 billion RNA sequences using our SSM task. Comprehensive evaluations demonstrate its SOTA performance in both RNA secondary and MSA-free tertiary structure prediction. Furthermore, our results highlight the benefits of the model's large parameter scale and the extensive pre-training dataset.

- **New PDB-derived Benchmark Dataset**: We construct and release a new and larger benchmark dataset for RNA secondary structure prediction, derived from the PDB. This dataset aims to facilitate more rigorous model evaluation.

## 2 RELATED WORKS

### 2.1 ENERGY-BASED FOLDING METHODS

Traditional RNA secondary structure prediction methods, like RNAstructure (Reuter & Mathews, 2010), CONTRAfold (Do et al., 2006) and others (Zuker, 2003; Lorenz et al., 2011b), use dynamic programming to find the structure with the Minimum Free Energy(MFE). They depend on thermodynamic parameters that come from experimental data. However, these parameters are limited. The experiments cannot capture all the details of RNA interactions under different biological conditions (Shi et al., 2014). MXfold2 (Sato et al., 2021) is a more recent method that uses deep learning to improve the energy parameters. It learns new parameters and then puts them into the dynamic programming framework.

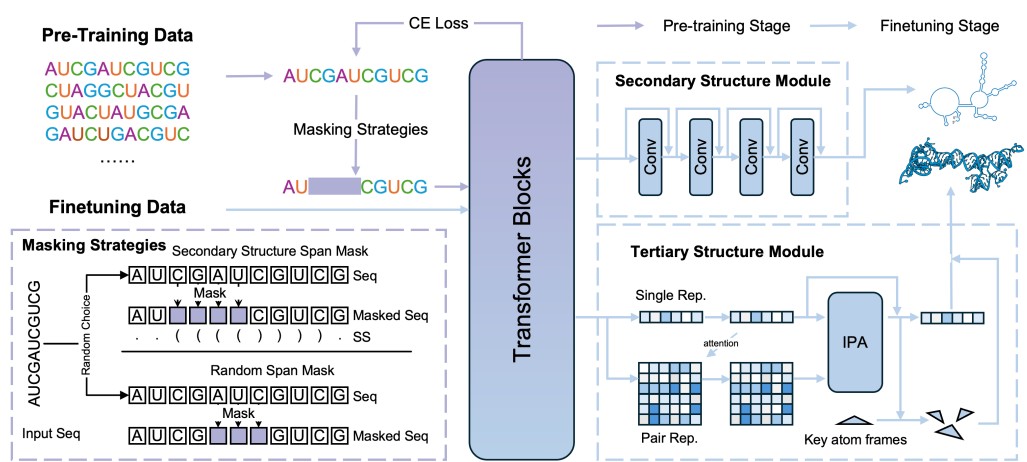

Figure 1: Overview of the SSR-LM model architecture. The model consists of a pre-training stage and a fine-tuning stage (blue and purple arrows, respectively). During the pre-training stage, diverse RNA sequences are masked by secondary structure-aware span masking or random span masking, and then the model learns to reconstruct the original sequences. The model incorporates a Secondary Structure Module and a Tertiary Structure Module to predict secondary or tertiary RNA structures.

## 2.2 DEEP LEARNING-BASED FOLDING METHODS

Recent progress in deep learning has led to new approaches for RNA secondary structure prediction. SPOT-RNA (Singh et al., 2019) is a method that uses deep learning for this task. It combines ResNet and bi-LSTM architectures with a sigmoid function. Other methods integrate deep learning with energy-based models. For example, MXfold (Akiyama et al., 2018), CDPfold (Zhang et al., 2019), and MXfold2 (Sato et al., 2021) all combine traditional models with deep learning techniques to improve predictions. E2Efold (Chen et al., 2020) learns unrolled algorithms to predict RNA structures and adds constraints to make the output valid. UFold (Fu et al., 2022) uses a U-Net architecture to further improve performance.

## 2.3 LANGUAGE MODEL-BASED FOLDING METHODS

Pre-trained language models have introduced novel avenues for RNA secondary structure prediction, primarily by leveraging self-supervised pre-training on extensive sequence datasets. RNA-FM (Chen et al., 2022) pioneered the application of PLMs to the secondary structure prediction task. ERNIE-RNA (Yin et al., 2024) significantly enhanced structural awareness by integrating base-pairing constraints, thereby achieving high F1 scores through the explicit modeling of complex interactions. RiNALMo (Penić et al., 2024b) highlighted the benefits of model scale, with its larger model size enabling superior performance and generalization over specialized deep learning methods on unseen RNA families.

## 3 SSR-LM

### 3.1 MODEL ARCHITECTURE

The architecture of SSR-LM is shown in Figure 1. Specifically, SSR-LM is based on the LLaMA2 encoder (Touvron et al., 2023), employing a stack of Transformer encoder blocks. We denote the number of layers as $L$, the hidden dimension size as $D$, and the number of Grouped Multi-Query Attention (GQA) heads as $H$. This work primarily reports results using one main model size: **SSR-LM** ($L$=33, $D$=1280, $H$=20, total Parameters≈650M). For ablation studies, we also developed two smaller models: **SSR-LM$_s$** ($L$=6, $D$=288, $H$=6, Total Parameters≈8M) and **SSR-LM$_m$** ($L$=12, $D$=1056, $H$=12, Total Parameters≈150M). Each Transformer encoder block consists of two RMSNorm layers, $H$ GQA heads, and a feedforward network that utilizes SwiGLU activation.

## 3.2 PRE-TRAINING SSR-LM

We pre-train SSR-LM using a masked language modeling objective, leveraging two distinct masking strategies detailed below. More details can be found in the Appendix A.

### 3.2.1 MASKING STRATEGIES

Let $X = (x_1, x_2, \ldots, x_n)$ be a sequence of input tokens, and $S = (s_1, s_2, \ldots, s_n)$ be its corresponding secondary structure sequence, where each $s_i \in \{`(`, `.`, `)`\}$.

**Span Masking(SM)**  This is the standard T5 (Raffel et al., 2020) masking strategy, serving as the baseline for our method. This strategy does not consider secondary structure information. We corrupt approximately 15% of the tokens in a sequence by randomly selecting spans of tokens. Each chosen span, $(x_i, \ldots, x_{i+j})$, where $j$ is a hyperparameter determining the span length offset, is replaced by a [MASK] token.

**Secondary Structure-Aware Span Masking(SSM)**  This strategy identifies tokens for masking based on the secondary structure sequence $S$. Let $\mathcal{Y}$ be the set of tokens selected for masking by SSM. The tokens in $\mathcal{Y}$ are identified as follows:

- If a token $s_i = `(`$: A span of tokens $(x_i, x_{i+1}, \ldots, x_k)$ is identified where all structural elements $s_p = `(`$ for $i \leq p \leq k$, and $s_{k+1} \neq `(`$ (or $k = n$). All tokens within this identified opening structural span are added to $\mathcal{Y}$.

- If a token $s_i = `)`$: A span of tokens $(x_k, \ldots, x_{i-1}, x_i)$ is identified where all structural elements $s_p = `)`$ for $k \leq p \leq i$, and $s_{k-1} \neq `)`$ (or $k = 1$). All tokens within this identified closing structural span are added to $\mathcal{Y}$.

- If a token $s_i = `.`$, the token $x_i$ will be back to SM mask.

All selected tokens in $\mathcal{Y}$ are replaced with the [MASK] symbol. SSM encourages the model to capture dependencies in structural base pairs by masking one side of each pair, while tokens outside paired regions are handled with SM.

### 3.2.2 PRE-TRAINING OBJECTIVE

During the pre-training process, each batch samples ncRNA and non-ncRNA from the pre-training dataset at a ratio of **1:1**. For ncRNA, there is a 50% probability of selecting secondary structure-aware span masking and a 50% probability of selecting span masking. For the non-ncRNA, span masking is applied. The model is trained to predict the original tokens for all masked positions using a standard masked language model objective. The loss function is defined as:

$$\mathcal{L}_{\text{MLM}} = - \sum_{k \in \mathcal{M}} \log P(x_k | X_{\backslash \mathcal{M}}), \tag{1}$$

where $\mathcal{M}$ is the set of indices of all masked tokens (selected either by SM or SSM), and $X_{\backslash \mathcal{M}}$ denotes the sequence of unmasked tokens.

### 3.2.3 PRE-TRAINING DATA

The pre-training dataset is sourced from the MARS database (Chen et al., 2024), a comprehensive RNA database containing 1.7 billion sequences, including RNAcentral, MG-RAST, Genome Warehouse, MGnify, and the NCBI nucleotide database and its associated subsets. For sequence standardization, uracil('U') was substituted with thymine('T') across all DNA sequences, thereby limiting the dataset to four tokens: 'A', 'C', 'G', and 'T'. Following this, we use MMSeqs2 to remove redundant sequences that shared 80% or greater similarity, reducing the dataset to **1.1 billion** sequences. This refined collection includes **3.1 million** ncRNA sequences, whose secondary structures were predicted by CONTRAfold (Do et al., 2006), are subsequently employed for the SSM task. A discussion on the choice of CONTRAfold for structure prediction is provided in the Appendix A.

### 3.3 FINE-TUNING SSR-LM

#### 3.3.1 UNSUPERVISED SECONDARY STRUCTURE PREDICTION

To assess the model's unsupervised learning capabilities for secondary structure prediction, we adapt the strategy proposed by ESMFold (Lin et al., 2023), which uses a logistic regression model to identify base pairing from attention maps. The probability of a contact $c_{ij}$ between nucleotides $i$ and $j$ is defined as:

$$p(c_{ij}) = \left( 1 + \exp\left( -\beta_0 - \sum_{l=1}^{L} \sum_{h=1}^{H} \beta_{hl} a_{ij}^{hl} \right) \right)^{-1}. \tag{2}$$

Here, $c_{ij}$ is a boolean random variable indicating whether nucleotides $i$ and $j$ are in contact. Our Transformer model has $L$ layers, each with $H$ attention heads. $A^{hl}$ denotes the symmetrized and Average Product Correction (APC)-corrected attention map for the $h$-th attention head in the $l$-th layer, and $a_{ij}^{hl}$ is the specific value from this map at position $(i, j)$. The parameters $\beta$ are fitted and do not backward the gradients through the attention weights.

#### 3.3.2 SUPERVISED SECONDARY STRUCTURE PREDICTION

For a consistent and fair comparison with RNA LM, we employ an identical ResNet network architecture as the supervised prediction head for secondary structure. Given an RNA sequence of length $L$, it is first processed by the pre-trained SSR-LM model to generate an $L \times D$ dimensional embedding. This embedding then serves as input to the ResNet network, which outputs an $L \times L$ matrix representing the predicted base-pairing probabilities for the secondary structure.

#### 3.3.3 TERTIARY STRUCTURE PREDICTION

RNA secondary structure provides the fundamental scaffold for its tertiary arrangement. Given this foundational role, and to further validate the quality of the embeddings learned by LM, we extend our experiments to tertiary structure prediction. We adapt the RhoFold (Shen et al., 2024) architecture to use embeddings from SSR-LM directly. First, the pre-trained SSR-LM model processes an RNA sequence to produce RNA embeddings. These embeddings are then used as input to the adapted RhoFold pipeline, replacing the original LM embeddings and completely excluding any MSA features. Throughout this process, the SSR-LM embeddings remain frozen. Within this framework, the structure module processes these embeddings to update both single and pair representations. Finally, the invariant point attention (IPA) module uses the updated representations to compute the rotation and translation matrices for each structural frame and then prediction the tertiary structure.

## 4 EXPERIMENTS

### 4.1 EXPERIMENTAL SETUP

#### 4.1.1 DATASETS

**Non-PDB-derived Datasets**   Our evaluation utilized two widely adopted non-PDB-derived datasets: ArchiveII and bpRNA. The ArchiveII dataset (Mathews, 2019) comprises 3,975 RNA sequences spanning 10 distinct classes. The bpRNA dataset (Danaee et al., 2018) contains 13,419 sequences, which are randomly split into training(10,814), validation(1,300), and test(1,305) sets.

**PDB-derived Datasets**   We build and evaluate our model on three datasets: PDB$_{2023}$, Atom-1 (Boyd et al., 2023), and CASP16. The PDB$_{2023}$ dataset is curated following the data processing methodology of SPOT-RNA (Singh et al., 2019), incorporating RNA structures deposited in the PDB prior to April 2023 (see Appendix A). This procedure yields 602 unique sequences (a nearly threefold increase compared to previous benchmarks), which were randomly split into training, validation, and test sets (424, 61, and 117 sequences, respectively). The Atom-1 dataset consists of 1,435 high-resolution monomeric RNA secondary structures, focusing exclusively on Watson-Crick (WC) base pairs and excluding regions with multiple junctions; its test set comprises 113 structures organized into 57 clusters, all published after May 1, 2020. Finally, the CASP16 dataset serves as a

Table 1: Performance comparison of SSR-LM with existing methods on secondary structure prediction tasks. The best results are highlighted in bold, and SSR-LM's results are highlighted in purple.

| | | ArchiveII | | | | bpRNA | | | |
|---|---|---|---|---|---|---|---|---|---|
| **Methods** | | Precision | Recall | F1 Score | AUPRC | Precision | Recall | F1 Score | AUPRC |
| | | | | | | | | | |
| | | *Non-PDB-devired Datasets* | | | | | | | |
| **Thermodynamics Method** | CONTRfold | 0.695 | 0.651 | 0.665 | 0.571 | 0.528 | 0.655 | 0.567 | 0.468 |
| **Deep learning Methods** | SPOT-RNA | 0.743 | 0.726 | 0.711 | 0.963 | 0.594 | 0.693 | 0.619 | 0.652 |
| | UFold | 0.887 | 0.928 | 0.905 | 0.931 | 0.521 | 0.588 | 0.553 | 0.398 |
| | RNA-FM | 0.912 | 0.876 | 0.894 | 0.927 | 0.763 | 0.623 | 0.685 | 0.686 |
| | ERNIE-RNA | 0.960 | 0.955 | 0.956 | 0.954 | 0.775 | 0.718 | 0.738 | 0.723 |
| | RiNALMo | 0.962 | 0.941 | 0.950 | 0.962 | 0.714 | 0.710 | 0.705 | 0.731 |
| **Ours** | SSR-LM | **0.966** | **0.958** | **0.960** | **0.973** | **0.820** | **0.702** | **0.757** | **0.778** |

| | | Atom-1 | | | | $PDB_{2023}$ | | | |
|---|---|---|---|---|---|---|---|---|---|
| **Methods** | | Precision | Recall | F1 Score | AUPRC | Precision | Recall | F1 Score | AUPRC |
| | | *PDB-devired Datasets* | | | | | | | |
| **Thermodynamics Method** | CONTRfold | 0.683 | 0.624 | 0.764 | 0.847 | 0.682 | 0.644 | 0.652 | 0.630 |
| **Deep learning Methods** | SPOT-RNA | 0.867 | 0.801 | 0.832 | 0.846 | 0.690 | 0.672 | 0.680 | 0.665 |
| | UFold | 0.830 | 0.760 | 0.789 | 0.687 | 0.703 | 0.554 | 0.612 | 0.461 |
| | RNA-FM | 0.833 | 0.762 | 0.796 | 0.810 | 0.819 | 0.575 | 0.676 | 0.654 |
| | ERNIE-RNA | 0.818 | 0.624 | 0.693 | 0.751 | 0.670 | 0.558 | 0.592 | 0.601 |
| | RiNALMo | 0.860 | 0.759 | 0.802 | 0.838 | 0.697 | 0.547 | 0.605 | 0.591 |
| **Ours** | SSR-LM | **0.889** | **0.805** | **0.845** | **0.861** | **0.823** | **0.674** | **0.741** | **0.721** |

blind test set. To ensure a meaningful and fair comparison, we excluded all homologous oligomeric targets with an F1 score close to 0; therefore, the final test set contains 9 targets (see Appendix A).

**Tertiary Structure Dataset** For training the tertiary structure prediction model, we utilized a distinct set of 5,024 RNA chains sourced from the PDB. To ensure a fair comparison, both SSR-LM and RNA-FM are trained for this task using identical experimental protocols and datasets. We curated a hard test set composed of 29 RNA structures from RNA-Puzzles (Bu et al., 2025) and six natural RNA targets from CASP15(see Appendix A). The training set for tertiary structure prediction is meticulously constructed to ensure no structural overlap with any of the defined test datasets.

### 4.1.2 IMPLEMENTATION DETAILS

All training details and evaluation metrics, including hyperparameter settings, optimizer configurations, and fine-tuning schedules, are provided in the Appendix A. For all supervised experiments involving PDB-derived datasets, model checkpoints evaluated were initialized with weights obtained from fine-tuning on the bpRNA dataset (Singh et al., 2019). For the unsupervised secondary structure prediction experiments conducted on $PDB_{2023}$, we use 20 RNA sequences for training, 20 for validation, and the remaining for testing (Lin et al., 2023). Evaluation metrics include Precision, Recall, F1 score, AUPRC, and lDDT.

### 4.2 SECONDARY STRUCTURE PREDICTION

### 4.2.1 STANDARD SECONDARY STRUCTURE PREDICTION

**Performance on Non-PDB-derived Datasets** We initially evaluate model performance on standard secondary structure prediction tasks using the ArchiveII and bpRNA datasets. As depicted in Table 1, SSR-LM consistently surpassed existing methods on the ArchiveII dataset. Specifically, SSR-LM achieves a leading F1 score of 0.960 and an AUPRC of 0.973, outperforming all other benchmarked models. For context, the next best-performing model, ERNIE-RNA, obtained an F1 score of 0.956. The superiority of SSR-LM is even more pronounced on the bpRNA dataset, where its AUPRC of 0.778 represented a 4.7 percentage point improvement over the secondary methods RINALMo.

**Performance on PDB-derived Datasets** Subsequently, we extend our evaluation to PDB-derived datasets. We think it is more important. As illustrated in Table 1, SSR-LM consistently demonstrates

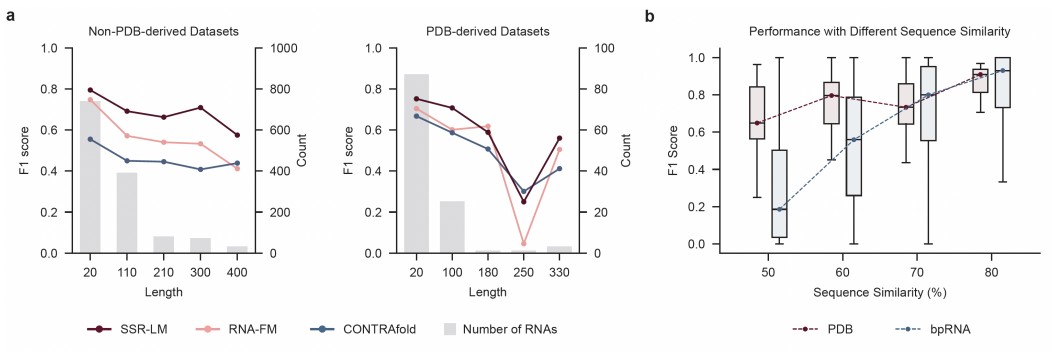

Figure 2: Secondary Structure Prediction Results Analysis. (a) Performance across varying sequence lengths. (b) Performance based on sequence similarity to the training set.

superior performance in these more challenging scenarios. On the Atom-1 dataset, SSR-LM achieved the highest F1 score of 0.845(+0.043). Similarly, on the more comprehensive PDB$_{2023}$ dataset, SSR-LM attained a leading F1 score of 0.741(+0.065). Interestingly, we observe that CONTRAfold, an energy-based method, exhibits a notable performance pattern. While it shows lower performance on non-PDB-derived datasets (0.567 on bpRNA), it achieves comparable F1 scores on PDB-derived datasets (0.680 on PDB$_{2023}$) to those of deep learning methods.

**Performance Analysis across Base-paring**   To further evaluate the specific strengths of SSR-LM, we conduct an analysis of predicted base-pair types. We chose two typical methods, RNA-FM and CONTRAfold, for comparison. We assess performance on Watson-Crick base pairs, wobble pairs, and non-canonical base pairs. As shown in Table S1, we find that SSR-LM not only excels in the prediction of standard WC(0.773) pairs and wobble pairs(0.453) but also performs well on NC pairs (0.302). This superior capability to discern a comprehensive spectrum of base-pairing interactions underscores the advanced predictive power and versatility of SSR-LM.

**Performance Across Sequence Lengths**   We investigate SSR-LM's adaptability to varying RNA sequence lengths (Figure 2a). On both the bpRNA and PDB$_{2023}$ datasets, F1 scores for most methods generally decrease as sequence length increases. SSR-LM maintains robust performance on bpRNA for sequences between 20–300 nucleotides (nt). On PDB$_{2023}$, while all methods perform well on shorter sequences, their F1 scores drop more significantly for RNAs exceeding 250 nt. In this challenging long-chain regime, SSR-LM shows superior adaptability and delivers markedly better predictive accuracy than RNA-FM.

**Performance Based on Sequence Similarity**   We examine SSR-LM's performance on sequences with varying similarity to its training set (Figure 2b). On the bpRNA dataset, F1 scores improve as sequence similarity increases. But this relationship is much weaker for the PDB dataset. Here, SSR-LM maintains strong F1 scores even with low training set similarity, and higher similarity does not yield noticeable improvements. This suggests sequence similarity provides little predictive benefit for the diverse, experimentally determined RNAs in the PDB dataset. Consequently, models rely more on their learned generalization abilities for such complex datasets.

### 4.2.2 GENERALIZATION EVALUATION

**Performance on Unseen CASP16 Targets**   To assess the model's generalization ability on novel and complex RNA structures, we evaluated its performance on the CASP16 benchmark. Since our training data is strictly limited to structures released before April 2023, the CASP16 dataset serves as a stringent temporal hold-out evaluation. As detailed in Table 2, SSR-LM achieves a SOTA F1 score of 0.62 and outperforms other methods. The superior performance on these challenging blind-test targets highlights SSR-LM's robust generalization capabilities and its potential for real-world RNA structure prediction applications.

Table 2: Casp16 Performance

| Methods | F1 score | AUPRC |
|---|---|---|
| CONTRAfold | 0.505 | 0.612 |
| RNA-FM | 0.496 | 0.462 |
| ERNIE-RNA | 0.416 | 0.412 |
| RiNALMo | 0.484 | 0.441 |
| SSR-LM | **0.635** | **0.645** |

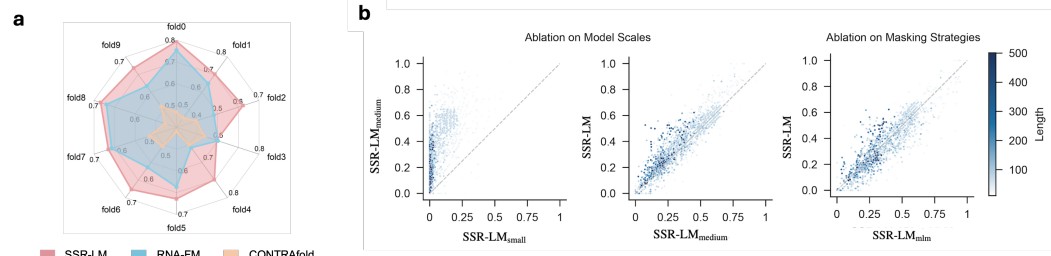

Figure 3: (a) Cross-family generalization performance. (b) Zero-shot performance between different model sizes and pretrain tasks.

**Cross-Family Generalization** We further benchmark cross-family performance using the classifications from Rfam, which organizes RNAs into 290 distinct families within our PDB$_{2023}$ dataset based on sequence and structure consensus (Kalvari et al., 2021). We perform a 10-fold cross-validation where, in each fold, a random subset of families constitutes the unseen test set. As detailed in Figure 3a, SSR-LM achieves an average F1 score of 0.679. This represents a substantial margin of +0.052 over the next-best method, RNA-FM (F1 score of 0.627), underscoring SSR-LM's strong ability to generalize to entirely novel RNA families.

### 4.3 TERTIARY STRUCTURE PREDICTION

**Performance on Tertiary Structure Prediction** On this task, SSR-LM demonstrates obvious improvements(Figure S1). It achieves an average lDDT of 0.55, outperforming the RNA-FM(0.51). Concurrently, for these same challenging tertiary structure targets, SSR-LM also exhibited superior accuracy in predicting their secondary structures, achieving an F1 score of 0.758, compared to RNA-FM's 0.734. These findings underscore SSR-LM's robust and comprehensive performance, excelling in both secondary and tertiary structure prediction aspects for complex RNA molecules.

**Reliability of Self-Estimated Confidence Scores** Furthermore, we evaluate the reliability of SSR-LM's self-estimated predicted lDDT (plDDT) scores, observing a strong Pearson correlation (r = 0.83) with the ground-truth lDDT (Figure 4a). This indicates that SSR-LM's confidence metrics are both trustworthy and informative.

**Performance Analysis based on Sequence Similarity** We further assess the generalization capabilities of SSR-LM by examining the relationship between achieved lDDT scores of the test targets and their sequence similarities to the training set. In this analysis, SSR-LM exhibited a Pearson correlation of 0.43 (Figure 4b). This correlation suggests that SSR-LM's predictions rely less on sequence homology, indicating superior generalization to novel RNA structures.

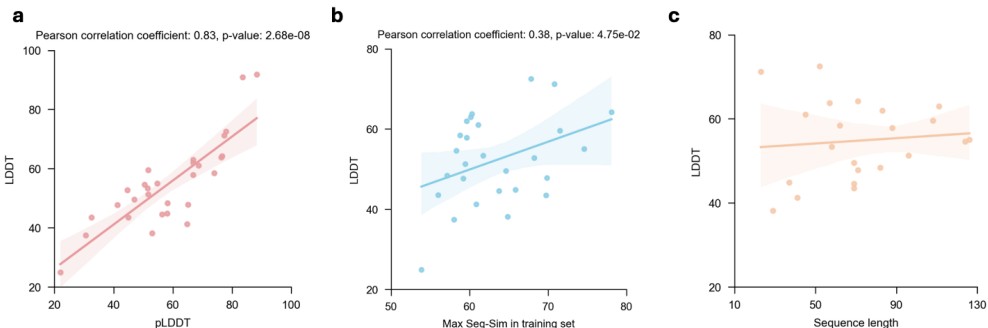

Figure 4: Correlation of predicted lDDT scores with ground-truth lDDT scores. (a) Correlation between predicted lDDT scores and ground-truth lDDT scores. (b) Correlation between achieved lDDT scores and sequence similarity to the training set. (c) Correlation between achieved lDDT scores and sequence length.

Table 3: Performance of Different Pretrain Tasks on Various Model Sizes

| Dataset | Pretrain Task | 650M | 150M | 8M |
|---------|---------------|------|------|-----|
| bpRNA | MLM | 0.736 | 0.697 | 0.652 |
| | SM | 0.742 | 0.712 | 0.622 |
| | SSM | **0.757** | **0.729** | **0.668** |
| $PDB_{2023}$ | MLM | 0.705 | 0.701 | 0.695 |
| | SM | 0.714 | 0.704 | 0.693 |
| | SSM | **0.741** | **0.722** | **0.707** |

**Performance Analysis based on Sequence Length**  SSR-LM exhibited consistent predictive accuracy across the spectrum of RNA lengths considered. This stability in performance is substantiated by a very low Pearson correlation coefficient ($r = 0.1$) observed between the sequence length and the resulting lDDT scores, as depicted in Figure 4c.

## 4.4 ABLATION

We conduct ablation studies to evaluate the impact of model scales and pre-training strategies on both unsupervised and supervised secondary structure prediction.

**Unsupervised Evaluation**  As illustrated in Figure 3b, we compare the unsupervised prediction performance between models of different sizes and models using different pre-training tasks. The results show that larger models consistently outperform smaller ones, with the majority of data points lying above the diagonal line. Similarly, our proposed SSM shows a clear advantage over the standard MLM task. A key observation is that for both comparisons, the data points for longer RNA sequences (indicated by darker blue) are skewed more significantly towards the top-left. This suggests that the performance gain from a larger model capacity and the SSM is particularly pronounced for longer RNAs, highlighting their superior ability to model complex long-range dependencies compared to smaller models or a simple MLM objective.

**Supervised Evaluation**  For the supervised setting, we benchmark different models on both bpRNA and $PDB_{2023}$. The results, summarized in Table 3, show that after fine-tuning, the SSM pre-trained models achieve the highest F1 scores across all model sizes (8M, 150M, 650M) on both datasets. This confirms the robust advantage of our pre-training strategy. Furthermore, we observe a notable trend on the PDB dataset: while the SSM and SM models benefit significantly from increased model size, the performance of the MLM-based model saturates, showing only marginal improvement as its size increases from 8M to 650M (0.695 to 0.705). This discrepancy suggests that for the intricate and often non-canonical structures in the PDB dataset, a generic task like MLM is insufficient for larger models to leverage their increased capacity. In contrast, the SSM task provides a more suitable inductive bias for RNA structure, enabling effective scaling and performance improvement.

**Latent Space Analysis of pre-training Strategies**  We analyze the latent space representations learned by SSR-LM through two pre-training tasks. For this, we use embeddings of 10,000 sequences from 27 families, covering four major RNA types (rRNA, snRNA, tRNA, and ribozyme) from the Rfam. Using UMAP to project these embeddings(Figure S2), we observe that models pre-trained with both SSM and MLM could distinguish between RNA types, but SSM pre-training results in much clearer clustering, particularly for ribozymes. More details are in Appendix A.

## 5 CONCLUSIONS

In this work, we introduce SSR-LM, an RNA language model featuring a novel pre-training task called secondary structure-aware span mask. Our experiments on both secondary and tertiary RNA structures validate that integrating structural information into the pertaining task can enhance structure prediction capabilities. As a result, SSR-LM achieves SOTA results and also shows strong generalization across RNA types and families. Our findings indicate that the performance benefits from the SSM task, model size, and pre-training data. In addition, we presented a new PDB-derived benchmark dataset. We hope that our work can inspire further exploration of RNA language models.

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

# A Appendix

## A.1 Experiment Setting Details

### A.1.1 Pre-training Details

**Dataset**   For the SSM of generating RNA secondary structures for our 3.1 million sequences, we employed CONTRAfold. This choice was based on several key considerations: (1) Our SSM task primarily focuses on masking stem regions. For these core structures, the predictive performance of most models, including the computationally efficient CONTRAfold, is largely comparable. (2) Computational resources were a critical factor. We estimated that using a state-of-the-art deep learning predictor would have required over 1500 GPU-hours for inference, which was impractical for our large-scale experiment. (3) CONTRAfold introduces a beneficial modeling bias. As a dynamic programming-based method, it guarantees that all predicted structures are physically valid (i.e., properly nested and without pseudoknots), ensuring a high-quality structural prior without the need for post-processing.

**Training Details**   In the pre-training stage, an equal proportion (1:1) of non-coding RNAs (ncRNAs) with predicted secondary structures and coding RNAs are sampled per batch. Subsequently, 50% of the ncRNAs undergo span masking as described by SpanBERT (Joshi et al., 2020), while the remaining ncRNAs and all coding RNAs are subjected to secondary structure span masking. Here, we set the SM length $j = 3$. This has been well-established in the T5 paper (Raffel et al., 2020). The batch size and maximum sequence length are both set at 1024, where we apply sequence packing. The SSR-LMmodel is trained over 175,000 iterations with a learning rate of $1 \times 10^{-4}$. SSR-LM is trained on 8 NVIDIA A100-80G GPUs for 7 days. We use the AdamW optimizer with a weight decay of 0.01.

### A.1.2 Fine-tuning Details

**Dataset**   For PDB dataset, we follow the steps of SPOT-RNA (Singh et al., 2019) to construct the PDB secondary structure. We downloaded 16,110 RNA structure data before April 2023 from the PDB website and used DSSR to extract their secondary structures. Different training subsets are then filtered by mmseq2 according to different minimum sequence identity cutoffs (80% and 100%). At a 100% identity cutoff, there are 2,029 unique sequences, of which 1,389 have annotated Rfam families and types. These are used for unsupervised secondary structure prediction and cross-family evaluation. We did not choose a stricter cutoff, as this would result in too few sequences in some categories. For supervised secondary structure prediction, we used the 80% identity cutoff, resulting in 602 sequences. For the CASP16 dataset, to ensure a meaningful and fair comparison, we excluded all homo-oligomer targets where all evaluated methods yielded an F1 score near 0; thus, the final test set includes targets R1209, R1211, R1242, R1261, R1262, R1263, R1283, and R1296.

**Unsupervised Secondary Structure Prediction**   We use same training target as ESMFold (Lin et al., 2023; Rao et al., 2020). To fit $\beta$, let $\mathcal{D}$ be a set of training proteins, $k$ be a minimum sequence separation, and $\lambda$ be a regularization weight. The objective can then be defined as follows:

$$\mathcal{L}(\mathcal{D}; \beta) = \prod_{d \in \mathcal{D}} \prod_{i=1}^{L_d - k} \prod_{j=i+k}^{L_d} p(c_{ij}^d; \beta) \tag{3}$$

$$\hat{\beta} = \max_{\beta} \mathcal{L}(\mathcal{D}; \beta) + \frac{1}{\lambda} \sum_{l=1}^{L} \sum_{h=1}^{H} |\beta_{lh}| \tag{4}$$

We fit the parameters $\beta$ via Pytorch and do not backpropagate the gradients through the attention weights. In total, our model learns $LH + 1$ parameters, many of which are zero thanks to the $L_1$ regularization. Same to ESMFold, all logistic regressions are trained with $|D| = 20$, $\lambda = 0.15$, k = 6.

**Supervised Secondary Structure Prediction**   For each dataset, SSR-LM train for 100 epochs with a batch size of 2 and a learning rate of $1 \times 10^{-4}$. The cosine learning rate schedule is employed with a warm-up period of 10 epochs. We utilize the AdamW optimizer with a weight decay of 0.01.

**Tertiary Structure Prediction**  The training dataset for RNA are from the PDB as of version 13-4-2022, including 13,379 chains. Chains exceeding 256 nucleotides or shorter than 16 nucleotides, as well as those with a resolution greater than 4 Å, are excluded. The Subsequent use of MMSeqs2 to remove sequences with greater than 80%. We selected 6 samples from CASP15 and 29 samples from the RNA-Puzzles (Cruz et al., 2012), which encompass experimentally determined RNA structures from RNA community challenges. Notably, there is no overlap between these evaluation samples and the training dataset. The SSR-LM-based tertiary structure prediction module was trained for 300,000 steps with a learning rate of $3 \times 10^{-4}$, utilizing 8 NVIDIA A100-80G GPUs.

### A.1.3 BASELINE IMPLEMENTATION DETAILS

**Checkpoints and Pre-trained Models**  For all baseline models, we utilized the officially released checkpoints provided by the original authors whenever they were available, ensuring our comparisons are based on the models as presented in their respective publications. However, some specific handling was required. We noted that the results reported in the RiNALMo cross-type evaluation paper were anomalous; therefore, we reproduced these specific experimental results. For the evaluation of ERNIE-RNA on the CASP16 results, we used their publicly released model weights trained on the RNA3D dataset and did not perform any retraining ourselves.

**Post-processing**  A critical step in our evaluation pipeline was the removal of all post-processing procedures for all baseline models. This ensures a fair and accurate calculation of the Area Under the Precision-Recall Curve (AUPRC), which can be skewed by model-specific heuristics. For RiNALMo, this specifically involved disabling `allow_flexible_pairings` while enabling `allow_sharp_loops` and `allow_nc_pairs`.

**Training Protocol**  In cases where a model required retraining, we followed a consistent and standardized procedure. We adopted the batch sizes reported in the original papers and performed a grid search for the optimal learning rate for each model across the set {5e-5, 1e-4, 5e-4}. The final model checkpoint selected for evaluation was the one that achieved the best F1 score on the designated validation set.

### A.1.4 EVALUATION METRICS

For unsupervised secondary structure prediction, we use Precision, which is same to ESMFold (Lin et al., 2023). For supervised secondary structure prediction, we employ the widely used F1 score as our primary evaluation metric. Because there are far fewer positive contacts than negative ones in RNA, we also report the area under the precision-recall curve(AUPRC) for a more comprehensive assessment. For tertiary structure prediction, we adopt the local Distance Difference Test (lDDT) score as our metric.

### A.2 ADDITIONAL RESULTS

**Performance Analysis across Base-paring**  To better understand SSR-LM's specific strengths, we analyzed its predictions across different base-pair categories and compared them with two representative methods, RNA-FM and CONTRAfold. We evaluated performance on Watson–Crick (WC), wobble, and non-canonical (NC) base pairs. As summarized in Table S1, SSR-LM achieves strong results on WC pairs (0.773) and wobble pairs (0.453), and also performs competitively on NC pairs (0.302). This ability to accurately capture a broad range of base-pairing interactions highlights the model's advanced predictive power and versatility.

Table S1: F1 Score for Different Models and Base Pairs

| Methods | WC pair | WOBBLE pair | NC pair |
|---|---|---|---|
| CONTRAfold | 0.675 | 0.336 | 0.129 |
| RNA-FM | 0.721 | 0.366 | 0.252 |
| SSR-LM | **0.773** | **0.453** | **0.302** |

**Performance on Tertiary Structure Prediction**  SSR-LM shows significant improvements in structure prediction (Figure S1). It achieves an average lDDT of 0.55 for tertiary structures, out-

performing RNA-FM's 0.51. For the same targets, SSR-LM also attains a higher F1 score of 0.758 in secondary structure prediction, compared to 0.734 for RNA-FM. This demonstrates SSR-LM's superior performance in predicting both secondary and tertiary structures for complex RNAs.

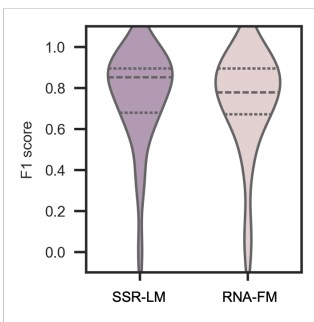

Figure S1: lddt on tertiary structure prediction dataset

**Reliability of Self-Estimated Confidence Scores** We assessed the reliability of SSR-LM's self-estimated confidence scores (plDDT). Our analysis reveals a strong Pearson correlation of 0.83 between the plDDT and the ground-truth lDDT scores (Figure 4a), indicating that the model's confidence scores are highly reliable.

**Latent Space Analysis of pre-training Strategies** We analyzed the latent space of embeddings from 10,000 RNA sequences across 27 families and four major types from Rfam. A UMAP projection (Figure S2) shows that while both SSM and MLM pre-training can distinguish RNA types, SSM provides significantly clearer clustering, especially for ribozymes. Furthermore, within each RNA type, the SSM-trained model effectively separated different families, a task where the MLM-trained model struggled, particularly with ribozymes and rRNA.

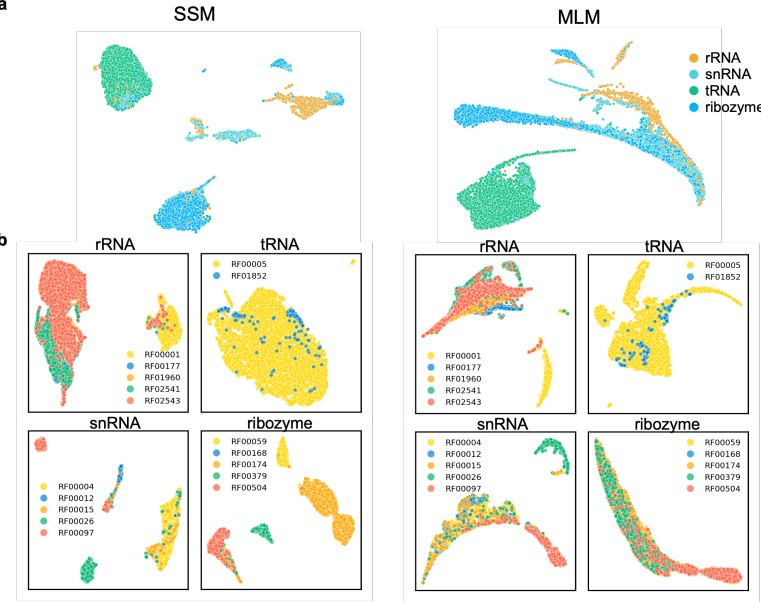

Figure S2: We collected 10,000 sequences from 27 families of the four most representative RNA types (rRNA, snRNA, tRNA, and ribozyme) from Rfam. These four types of RNA play different but very important roles in life activities. Using UMAP, we mapped the RNA embedding representations from different pretraining tasks into a two-dimensional plane to visually display the information encoded by SSR-LM.

### A.3 THE USE OF LARGE LANGUAGE MODELS (LLMS)

Large Language Models (LLMs) were employed as assistive tools in the preparation of this work. In particular, we used GPT-5 to support minor edits to academic writing, including drafting and refining sections. All scientific claims, methodological contributions, and experimental results were entirely conceived, implemented, and validated by the authors. The authors retain full responsibility for the content of this paper.

