# OpenReview forum: "Structured-Aware Pre-trained LM Can Improve RNA Structured Prediction"
_ICLR.cc/2026/Conference — ICLR 2026 Conference Withdrawn Submission_

### Official Review · Reviewer_zh2a · 2025-10-29

**Soundness:** 2
**Presentation:** 3
**Contribution:** 2
**Rating:** 4
**Confidence:** 4

**Summary:**

This paper introduces SSR-LM, a 650-million parameter RNA language model pretrained on a large dataset of 1.1 billion sequences. The authors propose a novel pretraining task, Secondary Structure-Aware Span Masking (SSM). This task uses secondary structures predicted by the CONTRAfold method to guide the masking strategy, with the goal of explicitly teaching the model long-range structural dependencies. The paper also introduces a new, larger PDB-derived benchmark dataset for evaluation. The authors report that SSR-LM achieves state-of-the-art performance on this new benchmark and on the CASP16 blind test set (0.635 F1-score).

**Strengths:**

1. The experiment results presented are comprehensive. The model achieves good performance across a wide array of secondary structure prediction tasks.


2. The authors contribute a new benchmark dataset derived from the PDB. This is a valuable resource for the community, as it addresses limitations of existing datasets and should allow for more rigorous model evaluation.


3. The evaluation is extended to MSA-free tertiary structure prediction. Showing that the SSR-LM embeddings lead to improved tertiary structure prediction provides strong additional evidence for the quality of the learned representations

**Weaknesses:**

1.  The pretraining for the SSM task relies on secondary structures predicted by CONTRAfold. The authors provide a reasonable justification for this choice, citing computational cost and the guarantee of valid, nested structures. This is understandable. However, one might wonder about the quality of these structural labels. If the predicted structures from CONTRAfold contain significant inaccuracies, this could introduce noise into the pretraining task and perhaps limit the model's ultimate performance.

2.  The paper could elaborate more on the precise mechanism of the SSM task. The current description explains that spans of opening or closing brackets are masked. It is not perfectly clear how this handles complex, non-contiguous pairings, or if the masking is based only on contiguous stem regions. A more detailed example would be beneficial. Additionally, the pretraining strategy mixes SSM with standard Span Masking for ncRNA sequences. The paper could offer more justification for this specific 50/50 ratio and explain why this mixture is superior to using SSM alone on all ncRNA data.

3.  The core idea of using predicted structural information to guide a masking task is quite intuitive. While the SSM task is clearly effective and well implemented, the conceptual novelty of this approach might be viewed as an incremental improvement rather than a major methodological leap. The paper's contribution is very strong empirically, but the novelty of the pretraining task itself is perhaps more modest.

**Questions:**

1. The justification for using CONTRAfold is clear regarding computation15. However, what is your assessment of the quality of these predicted structures, especially for complex or non-canonical RNAs? Have you analyzed the potential impact of noisy or inaccurate structural labels from CONTRAfold on the pretraining process?

2. Could you provide a more detailed example of how SSM operates? Does this only apply to contiguous stems? How would it handle a long-range pairing where the opening ( at position 5 and closing ) at position 100 are separated by a large internal loop? Would it mask position 5 and position 100, or just one of them?

3. For ncRNA, you apply SSM 50% of the time and standard Span Masking (SM) 50% of the time. What was the rationale for this 50/50 split? Did you experiment with other ratios, or use SSM 100% of the time for ncRNA? Is there a risk of the two different masking strategies sending conflicting signals to the model during pretraining?

---

### Official Review · Reviewer_Mrpd · 2025-10-29

**Soundness:** 2
**Presentation:** 3
**Contribution:** 3
**Rating:** 4
**Confidence:** 4

**Summary:**

The paper proposes a new RNA Language Model, SSR-LM, that was trained across 1.1 billion sequences using a novel pre-training strategy, secondary structure aware span masking (SSM). The SSM approach masks sequences based on secondary structure features and was used during training on a subset of ~3 million datapoints of non-coding RNAs with secondary structures predicted by ContraFold. The model is evaluated for supervised prediction performance on 5 secondary structure datasets (ArchiveII, bpRNA, a new PDB testset, Atom-1 testset, and a subset of the CASP16 RNA targets)  against several baselines and for 3D structure prediction on CASP 15 rna targets against RNA-FM. The authors further analyze different pre-training strategies across model sizes in an ablation study and propose a new benchmark dataset derived from PDB.

**Strengths:**

- The new pre-training strategy is interesting and seems to work well.
- From the reported results, SSR-LM outperforms the other methods across several datasets and tasks.
- The authors provide substantial analysis, including performance analysis of different base-pair types, sequence lengths, and sequence similarity, as well as an ablation study and latent-space analysis.

**Weaknesses:**

**Major**:
1. The new PDB dataset is one of the three major contributions. However, it is curated following a pipeline of [1] which was already raised concerns in the secondary structure community because data splitting is based solely on sequence similarity (see e.g. [2,3]). Although larger than the dataset used in [1], I think this dataset should not become the new standard for DL methods in the community. Why not process RNA3DB [4] to extract secondary structures which was curated based on structure / family similarity? Since the authors already process PDB files, that should be a straightforward approach for a much cleaner data setup.
2. The authors already mention that it is less important than evaluations on PDB data, but the bpRNA dataset suffers from similar data problems described in 1. (see [5]).
3. Due to the data issues mentioned in 1., one could argue that the strong performance reported for supervised secondary structure prediction mainly results from learning similarities between train and test data.
4. For tertiary structure predictions, the authors claim that there is no structural overlap between the datapoints, however, in the appendix, I can only see sequence similarity measures, or? Then, I see similar data issues for the CASP15 and RNA-Puzzles evaluations. Again, I would encourage the authors to either use RNA3DB or a data processing pipeline that ensures splitting based on structure similarity (this could e.g. be done via TM-Score; see e.g. [6] supp. material)
5. Since all supervised experiments were initialized using bpRNA weights, there could be even more overlap between the data I guess.

**Minor**:
1. In table 1, results for SSR-LM are accidentally bold for recall on bpRNA while ERNIE is better; Pearson correlation score in Section 4.3 analysis on seq. Similarity is different in test and figure.
2. Regarding cross-family evaluations for secondary structures, how were families assigned? Using Infernal? Which e-value?
3. It would be interesting to see the performance of the vanilla RhoFold on the tertiary structure targets for comparison.


[1] Singh, J., Hanson, J., Paliwal, K., & Zhou, Y. (2019). RNA secondary structure prediction using an ensemble of two-dimensional deep neural networks and transfer learning. Nature communications, 10(1), 5407.

[2] Szikszai, Marcell, et al. "Deep learning models for RNA secondary structure prediction (probably) do not generalize across families." Bioinformatics 38.16 (2022): 3892-3899.

[3] Qiu, X. (2023). Sequence similarity governs generalizability of de novo deep learning models for RNA secondary structure prediction. PLOS Computational Biology, 19(4), e1011047.

[4] Szikszai, M., Magnus, M., Sanghi, S., Kadyan, S., Bouatta, N., & Rivas, E. (2024). RNA3DB: A structurally-dissimilar dataset split for training and benchmarking deep learning models for RNA structure prediction. Journal of Molecular Biology, 436(17), 168552.

[5] Flamm, Christoph, et al. "Caveats to deep learning approaches to RNA secondary structure prediction." Frontiers in Bioinformatics 2 (2022): 835422.

[6] Tan, C., Zhang, Y., Gao, Z., Cao, H., Li, S., Ma, S., ... & Li, S. Z. (2025). R3Design: deep tertiary structure-based RNA sequence design and beyond. Briefings in Bioinformatics, 26(1), bbae682.

**Questions:**

1. I might have overlooked it but how was SSR-LM fine-tuned for evaluations on ArchiveII?
2. Is there a rationale beyond compute limitations why the SSM was only applied to the ncRNA training samples?
3. It seems that the SSM mainly masks stem regions. Would the authors think that explicitly masking loop regions (e.g. GNRA-tetra loops are known to be important) could be beneficial for model pre-training?

---

### Official Review · Reviewer_8kkL · 2025-10-31

**Soundness:** 3
**Presentation:** 2
**Contribution:** 2
**Rating:** 2
**Confidence:** 5

**Summary:**

This paper proposes a new RNA language model, SSR-LM, which was pretrained using a secondary structure-aware pretraining task. During the pretraining task, sequence tokens are masked based on the secondary structure, such that a contiguous span of nucleotides within one strand of a structural element is masked, e.g., a stem. Such a pretraining approach provided an inductive bias for secondary structure prediction tasks. The model size is similar to the concurrent model sizes and was pretrained on both non-coding and coding RNAs from the MARS dataset. The model was evaluated only on non-coding structure prediction tasks, where it showed the benefits of a structure-aware pretraining task.

**Strengths:**

- The paper proposed a novel structure-aware pretraining task, SSM, where the model masks sequences based on important structural spans and learns to recover them. The authors demonstrated the benefits of using this pretraining approach on several structure-related downstream tasks. I find it as an interesting way of providing an inductive bias for structure-related downstream tasks.
- The authors constructed a new PDB-derived secondary-structure benchmark dataset using the DSSR tool. While I praise the idea of building new benchmark datasets, following the steps of SPOT-RNA for data processing and curation is not the best and most strict way of creating an RNA evaluation benchmark. Additionally, the authors were not clear whether they did additional data curation to remove potential sequence homologies with the bpRNA training dataset.
- The authors did ablation study on model sizes and pretraining tasks. It was clearly shown that larger models perform better when evaluated on unsupervised stucture prediction tasks. Additionally, the ablation shows benefits of the SSM pretraining task over the traditional MLM pretraining task when evaluated on unsupervised stucture prediction.

**Weaknesses:**

**Major comments:**
- The benefits of pretraining the model on 1.1B coding and non-coding RNA (ncRNA) sequences are not clear. The authors claim that, after data curation, the pretraining dataset is left with only 3.1M ncRNAs. Having such a small fraction of ncRNAs is really strange. Since all the downstream tasks are ncRNA structure-related, the authors should explain the benefits of coding RNAs in the pretraining dataset and why they didn't employ bigger ncRNA-only datasets, e.g., RNAcentral, Rfam or NCBI's NT.
- It's not clear how the proposed pretraining task influences the model's performance on other function-related tasks. The paper presented only secondary structure prediction results and a single tertiary structure prediction task. I suggest evaluating the model on function-related tasks as well to further assess the benefits of SSM.
- In the paper, CONTRAfold is utilized for the secondary structure prediction of 3.1M ncRNAs. CONTRAfold is far from a perfect secondary structure prediction tool. The authors should be aware and comment on the biases induced using CONTRAfold and not the real secondary structures.
- **Latent space analysis:** The paper analyzed the latent space of embeddings from 10,000 ncRNAs across 27 Rfam families. The paper asserts that SSM provides much clearer clusters, but this is hardly visible from the figures. Except for ribozymes, the model hardly classifies different Rfam families.
- **Secondary structure prediction:**
  - In this paper, ArchiveII and bpRNA datasets are split using only sequence similarity. It was shown in several previous papers, such as RNA-FM. RiNALMo, MXfold2, etc., that deep learning models perform excellently on in-distribution data, but struggle on unseen RNA families. I suggest evaluating SSR-LM on proper secondary structure prediction tasks. If utilizing the ArchiveII dataset, the model can be tested using cross-validation, such as proposed in [1]. Additionally, the TORNADO dataset [2] was created by splitting data based on structural similarity and provides a much harder structure prediction task. I suggest using these datasets instead of the current ones.
  - The way the PDB_2023 dataset was constructed again does not account for structural similarity between the training and test datasets. Additionally, the paper states that for the supervised experiments involving PDB_2023, SSR-LM was initialized with weights obtained from fine-tuning on the bpRNA dataset. The way PDB_2023 was constructed, it is not clear whether structurally similar sequences from bpRNA were excluded from the test set and whether sequence homologs were removed.
  - The paper stated SSR-LM maintains robust performance on different sequence lengths; however, it is clear from Figure 2a that the performance decreases with the length, and this is especially clear for the PDB-derived datasets.
  - Figure 2b shows performance with respect to different sequence similarity; however, for the PDB dataset, the initial weights were taken from the bpRNA dataset. Are the reported sequence similarities exclusively for the PDB dataset, or include bpRNA and PDB together?
  - In several cases, for example, Figure 2a and Figure 3a, RNA-FM is used as a baseline. It was shown that larger models perform better on structure prediction tasks, capturing long-range dependencies better. I would expect to see a comparison with the same size model, e.g., RiNALMo or AIDO.RNA, to clearly see the benefits of pretraining on bigger data and using the proposed pre-trained task.
- **Tertiary structure prediction:**
  - SSR-LM, a 650M parameter model, was compared to RNA-FM, which is a 100M parameter model. It is not clear where the benefits come from; is it from the model size? How important are these results when compared to other tertiary structure prediction models? The only reported result is a comparison to RNA-FM. Also, the results are very close to each other.
- **Reproducibility:**
  - The paper does not state whether the model weights and pretraining/finetuning code will be made available. This will make reproducing the results and comparing on the PDB_2023 dataset and other tasks using a pretrained model on the PDB_2023 dataset impossible.

**Minor comments:**
- The paper claims that the model was pretrained on 1.1B sequences from the MARS dataset. Being pretrained for 175,000 iterations and having half of the 1024 batch size dedicated to only 3.1M ncRNA sequences, the model does not see all of the sequences. This should be clearly stated in the paper.
- In line 374, the authors report a different result than the one in Table 2.

---

[1] Szikszai et al., Deep learning models for RNA secondary structure prediction (probably) do not generalize across families. Bioinformatics. 2022.

[2] Rivas et al., A range of complex probabilistic models for RNA secondary structure prediction that includes the nearest-neighbor model and more. RNA. 2011.

---

Overall, I recommend **rejecting** the paper for the following two main reasons:
1. Dataset composition and pretraining rationale are unclear. Pretraining on 1.1B sequences that yield only 3.1M ncRNAs after curation needs a clear justification. Why include coding RNAs, and why not use larger ncRNA resources (RNAcentral, Rfam, NCBI NT)?
2. The way the model is evaluated and compared in the current version of the paper is not sufficient. Including other function-related downstream tasks and proper secondary structure evaluation datasets would help in evaluating the model properly.

The manuscript proposes an interesting structure-aware pretraining approach for RNA modeling, but in its current form, the experiments and reporting do not convincingly demonstrate the claimed benefits and suffer from potential data/methodological confounds.

**Questions:**

1. Could the authors explain the benefits of coding RNAs in the pretraining dataset and why they didn't employ bigger ncRNA-only datasets, e.g., RNAcentral, Rfam, or NCBI's NT? Please comment on only 3.1M ncRNAs.
2. Please comment on the limitations of using CONTRAfold for predicting large-scale ground-truth structure proxies.
3. Could you explain in more detail the creation of the PDB_2023 dataset and whether structurally similar sequences from bpRNA were excluded from the test se,t and whether sequence homologs were removed?
4. In Figure 2b, are the reported sequence similarities exclusively for the PDB dataset, or include bpRNA and PDB together?
5. For tertiary structure prediction, it is not clear where the benefits come from; is it from the model size? How important are these results when compared to other tertiary structure prediction models?
6. Will the authors release model weights and pretraining/finetuning code upon acceptance of the paper?
7. Please comment on the number of coding sequences seen during pretraining.

---

### Official Review · Reviewer_fdYA · 2025-11-01

**Soundness:** 2
**Presentation:** 3
**Contribution:** 2
**Rating:** 4
**Confidence:** 4

**Summary:**

The paper proposes an RNA LM trained with a secondary-structure–aware span masking (SSM) objective. The idea is clearly motivated, the masking rule is well specified and reproducible, and the evaluation spans secondary (supervised/unsupervised) and tertiary tasks with a frozen-embedding pipeline. The newly assembled PDB-derived benchmark also has potential community value.

**Strengths:**

Introduces one-sided stem span masking directly into the pretraining objective; the motivation is clear and the procedure is reproducible.

Evaluates across secondary structure (both supervised and unsupervised) and tertiary structure using frozen embeddings, supporting the claim that the learned representations transfer.


Assembles a larger PDB-derived benchmark and reports scaling trends across multiple model sizes, which is valuable for the community.

**Weaknesses:**

Incorporating RNA secondary structure in pretraining is not new. The paper does not clearly articulate how the proposed approach compares to prior structure-aware methods

No discussion of pseudoknots. The manuscript omits analysis of pseudoknot handling, including whether the approach inherits biases from predictors that cannot model pseudoknots, and how this affects performance on pseudoknot-rich RNAs.

**Questions:**

1. The non-PDB side reports ArchiveII/bpRNA but omits BPnew, a stricter split commonly used to test generalization beyond bpRNA. This weakens the generalization claim on non-PDB benchmarks.

2. Injecting RNA secondary structure during pretraining is not new (e.g., ERNIE-RNA, RiNALMo, structRFM). What remains unclear is why this specific SSM mechanism helps beyond learning CONTRAfold’s biases, since SSM is driven by CONTRAfold-predicted brackets at scale. It is plausible the model partially mimics CONTRAfold rather than learning a tool-agnostic structural prior, raising data/label leakage concerns if the pretraining exposure aligns with evaluation labels/distributions.

3. On page 6, Precision = 0.683, Recall = 0.624, but F1 = 0.764 for Atom-1/CONTRAfold, impossible because F1 must be ≤ max(P, R). This undermines confidence and suggests a broader table audit is needed.

---

### Note · Authors · 2025-12-04

I have read and agree with the venue's withdrawal policy on behalf of myself and my co-authors.